

# Comprehensive aerosol and gas data set from the Sydney Particle Study

Melita Keywood[1], Paul Selleck[1], Fabienne Reisen[1], David Cohen[2], Scott Chambers[2], Min Cheng[1], Martin Cope[1], Suzanne Crumeyrolle[3],Erin Dunne[1], Kathryn Emmerson[1], Rosemary Fedele[4] , Ian Galbally[1], Rob Gillett[1], Alan Griffiths[2], Elise-Andree Guerette[1,5], James Harnwell[1], Ruhi Humphries[1], Sarah Lawson[1], Branka Miljevic[6] Suzie Molloy[1], Jennifer Powell[1], Jack Simmons [5], Zoran Ristovski[6], Jason Ward[1]

[1]CSIRO Oceans and Atmosphere, PMB1 Aspendale, VIC 3195, Australia
[2] ANSTO, Environmental Research, Locked Bag 2001, Kirrawee DC, NSW 2232, Australia
[3]Univ. Lille, CNRS, UMR 8518 - LOA - Laboratoire d'Optique Atmosphérique, 59000 Lille, France
[4] EPA Victoria Melbourne Vic 3001 Australia
[5] University of Wollonong School of Chemistry University of Wollongong N.S.W. 2522 Australia
[6] School of Chemistry, Physics and Mechanical Engineering, Queensland University of Technology Brisbane QLD 4001 Australia

Correspondence to: Melita Keywood (melita.keywood@csiro.au)

**Abstract.**

The Sydney Particle Study involved the comprehensive measurement of meteorology, particles and gases at a location in western Sydney during February/March 2011 and April/May 2012. The aim of this study was to increase scientific understanding of particle formation and transformations in the Sydney airshed. In this paper we describe the methods used to collect and analyse particle and gaseous samples, as well as the methods employed for the continuous measurement of particle concentrations, particle microphysical properties and gaseous concentrations. This paper also provides a description of the data collected and is a meta data record for the data sets published in Keywood et al. (2016a) http://doi.org/10.4225/08/57903B83D6A5D and Keywood et al. (2016b) http://doi.org/10.4225/08/5791B5528BD63.

## 1. Introduction

Atmospheric particles adversely effect human health, impacting mortality and morbidity ((Pope et al., 2002), and are a significant contributer to outdoor air pollution being recognised by the World Health Organisaiton as carcinogenic to humans (Lim et al., 2012). Atmospheric particles are derived from a wide range of natural and anthropogenic sources, and hence are made up of a range of sizes and chemical compostions. This makes reduction of particle concentrations in the atmosphere by source regulation very challenging. In particular, reduction of secondary particles, which can be an important component of total particle exposure (Brook et al., 2010), are generated by photochemical reactions in the atmosphere and hence require control mechanisms that consider the relevant gas-phase precursors to these particles.



In the most recent Australian State of the Environment report, air quality standards were most often exceeded for fine particles in the capital cities, whilst ozone and nitrogen dioxide standards were not exceeded (Keywood et al., 2017). Currently, the highest episodes of particle pollution in Sydney can be ascribed to the presence of bushfire and dust plumes in the Sydney airshed (e.g. Johnston et al., 2011). However, significant increases in the the frequency of hot days, drought and high fire risk

weather have been projected for New South Wales, Australia (Whetton et al., 2015). The increased frequency of hot and sunny days has been linked to photochemical smog severity (Schnell and Prather 2017). Thus projected warmer conditions re likely to have implications air pollution and health in NSW.

Comprehensive chemical transport modelling tools can be used to assist in the development of a long term control strategy for particles in the Sydney airshed. Such models should encompass comprehensive three-dimensional simulations of the

atmosphere, sources and multi-phase chemistry that occurs and should be informed by understanding of the contribution made by both local and remote particle sources to total particle exposure within the region. Ultimately such understading should be underpinned by detailed and high quality experimental studies.

The Sydney Particle Study (SPS) aimed to increase scientific knowledge of the processes leading to particle formation and transformations in Sydney through two comprehensive observation programs. The groups that contributed to these observation

programs included CSIRO, NSW Office of Environment and Heritage, ANSTO, Queensland University of Technology, the Shanghai Institute of Applied Physics and University of Wollongong. Observation made included the collection of samples for chemical analysis (size-resolved aerosol composition, speciated volatile organic compounds [VOCs] including alkanes, aromatics, carbonyls, isoprene and monoterpenes, acid gases). In addition, continuous or semi-continuous measurements of aerosol number size distributions, aerosol mass, aerosol light scattering, aerosol composition, and gaseous criteria pollutants:

oxides of nitrogen, carbon monoxide, sulfur dioxide and ozone [$NO_X$, $CO$, $SO_2$, $O_3$]). Measurements were also made of meteorological parameters (wind speed, wind direction, temperature, relative humidity, radiation, boundary layer height) and atmospheric Radon-222 (radon) concentration.

## 2 Measurement Site

Measurements were made at the Westmead air quality station operated by the New South Wales Office of Environment and

Heritage, located 24 km to the west of the Sydney, Australia. The population of Sydney is 4.61 million in 2011 (ABS 2011), making Sydney the largest urban centre in Australia. Sydney is a coastal city with coastline to its east and elevated forested terrain (up to 1000 m) to the north, west and south and the climate is temperate with uniform rainfall, warm summers and cool winters.

The SPS observations occurred in two time periods; Summer 2011 (5 February - 7 March 2011 SPS-I) and autumn 2012 (16

April-14 May 2012 SPS-II).



## 3 Instruments and methods

The sampling program included the measurement of aerosols, criteria gases including NOx, CO, $SO_2$ and ozone, acid/alkaline gases including $NH_3$, $SO_2$, HCl and $HNO_3$, speciated VOCs (including carbonyls), and meteorological parameters, including temperature, relative humidity (RH) and wind speed/direction and boundary layer height. Aerosols were measured with

continuous or semi-continuous methods, included in the measurement of aerosol mass, light scattering and number size distributions as well as integrated measurements of aerosol composition. Atmospheric radon concentration were also measured and provided as an indicator of transport and vertical mixing processes as described in Scott et al. (2019).

Two integrated samples (particles, VOCs and acid/alkaline gas) were collected each day (morning 05:00 to 10:00 and afternoon 11:00 -19:00). Note that these times were local time (GMT+11 for SPS-I, GMT+10 for SPS-II). In addition, third a VOCs

(adsorbent tube/DNPH sampling) sample was collected between 19:00 and 05:00 (i.e. overnight). Table 1 summarises the parameter measured, the instrumentation used and frequency of the measurement for both SPS-I and SPS-II.

**Table 1. Measurements made at Westmead during SPS-I and SPS-II along with the instrument or analytical technique employed, the measurement and reporting resolution, and the measurement units.**

| Parameter | Instrument/ Analysis technique | Resolution | Reported resolution | Units | Period |
|---|---|---|---|---|---|
| Number Size distribution 3-150 nm | Scanning mobility particle sizer (SMPS- Nano) with TSI 3085 DMA column and TSI 3776 Condensation Particle Counter (CPC) | 5 min | 5 min | dN/dLogdp particles $cm^{-3}$ | SPS-I |
| Number Size distribution 15 – 750 nm | SMPS-Long with TSI 3071A DMA and TSI 3010 CPC | 5 min | 5 min | dN/dLogdp particles $cm^{-3}$ | SPS-I |
| Number Size distribution 15 – 750 nm | SMPS-Long with TSI 3081 DMA ,TSI 3010 CPC and TSI controller (3080). | 2.5 min | 2.5 min | dN/dLogdp particles $cm^{-3}$ | SPS-II |



| Parameter | Instrument/ Analysis technique | Resolution | Reported resolution | Units | Period |
|---|---|---|---|---|---|
| Total particle number concentration | CPC TSI 3772 | continuous | 1 minute | particles cm$^{-3}$ | SPS-I & SPS-II |
| PM$_{2.5}$, OC/EC, sugars (incl. Levoglucosan), water soluble ions | PM$_{2.5}$ Ecotech 3000 high volume sampler / DRI Model 2001A Thermal-Optical Carbon Analyzer/Ion Chromatography | Integrated (2 samples per day) on all days | 05:00-10:00, 11:00-19:00 | µg m$^{-3}$ | SPS-I & SPS-II |
| PM$_{2.5}$ elemental analysis | PM$_{2.5}$ ASP Sampler/ Ion beam analysis ANSTO STAR 2MV accelerator | Integrated (2 samples per day) on all days | 05:00-10:00, 11:00-19:00 | µg m$^{-3}$ | SPS-II |
| TSP mass | RAAS | Integrated (2 samples perday) on all days | 05:00-10:00, 11:00-19:00 | µg m$^{-3}$ | SPS-I |
| Light scattering | Nephelometer Ecotech 1000G | Continuous | hourly | mM$^{-1}$ | SPS-I & SPS-II |
| PM$_{10}$ | Thermo TEOM 1405 | Continuous | hourly | µg m$^{-3}$ | SPS-I & SPS-II |
| Radon | 700 L dual flow-loop two-filter radon detector | Continuous | 30 min and hourly | Bq m$^{-3}$ | SPS-I & SPS-II |
| CO | Ecotech EC9830 | Continuous | hourly | ppb | SPS-II |



| Parameter | Instrument/ Analysis technique | Resolution | Reported resolution | Units | Period |
|---|---|---|---|---|---|
| NO, NO$_2$, NO$_y$ | Ecotech EC9841 | Continuous | hourly | ppb | SPS-I & SPS-II |
| Ozone | Ecotech EC9810 | Continuous | hourly | ppb | SPS-I & SPS-II |
| SO$_2$ | Ecotech EC9850 | Continuous | hourly | ppb | SPS-I & SPS-II |
| NH$_3$, SO$_2$, HNO$_3$ | Gas filter sampler/ Ion Chromatography | Integrated (2 samples/day) on all days | 05:00-10:00, 11:00-19:00 | ppb | SPS-I & SPS-II |
| VOCs | Proton transfer reaction mass spectrometry (PTR-MS) | Continuous | hourly | ppb | SPS-I & SPS-II |
| VOCs | adsorbent tube/GCMS | Integrated - 3 samples/ day on all days | 05:00-10:00, 11:00-19:00, 19:00-05:00 | ppb | SPS-I & SPS-II |
| Carbonyls | S10 DNPH sampling/HPLC | Integrated - 3 samples/ day on all days | 05:00-10:00, 11:00-19:00, 19:00-05:00 | ppb | SPS-I & SPS-II |
| Wind speed & wind direction | Met-One MET505 G4056 | Continuous | hourly | m s$^{-1}$ & ° | SPS-I & SPS-II |
| Temperature & humidity | Vaisala HMP 155 | Continuous | hourly | °C & % | SPS-I & SPS-II |
| Solar | Middleton 8536 | Continuous | hourly | W m$^{-2}$ | SPS-I & SPS-II |
| Boundary layer height | Leosphere ALS 450 lidar | 30 s | 20 min | m | SPS-II |


### 3.1 Continuous and semi-continuous measurements

### 3.1.1.Aerosol microphysical measurements

Aerosol size distributions were measured by different instruments during SPS-I and SPS-II. During SPS-I two instruments were used: a Scanning Mobility Particle Sizer (SMPS) which was custom built and included a long Differential Mobility

Analyser (DMA, TSI 3071A) column and CPC (TSI 3010) (Long-SMPS) and a nano-SMPS which was also custom built and consisted of a short DMA (TSI 3085) column and CPC (TSI 3776) (Nano-SPMS). Both the Long-SMPS and Nano-SMPS



were run with aerosol flows of 0.30 ± 0.03 L min⁻¹ and sheath flows of 3.0 ± 0.3 L min⁻¹ resulting in distribution of particles between 15 - 736 nm being measured with the Long-SMPS and the distribution of particles between 4.6-156 nm being measured with the Nano-SMPS. Size distribution scans occurred over 5 minute intervals and PolyStyrene Latex (PSL) spheres

were used to determine the sizing accuracy of both SMPS systems (± 2 %).

Comparison of the total number concentrations of particles greater than 10 nm measured with the Nano-SMPS to the particle number concentration measured using the CPC TSI3772 determined the counting efficiency of the Nano-SMPS and a scaling factor was determined which was then used to scale the Nano-SMPS size distributions. The Nano-SMPS and Long-SMPS had an overlap between 15 and 156 nm.  The relationship between the concentrations measured in the overlapping size ranges was

used to scale the Long-SMPS concentrations to the Nano-SMPS concentrations. Merging of the Long-SMPS and Nano-SMPS data sets produced a distribution between 4.6 nm to 736 nm.

During SPS-II aerosol size distributions were measured using an SMPS which included a long DMA column(DMA, TSI 3081) column and CPC (TSI 3010) and the TSI controller (3080). The SMPS was run with aerosol flows of 0.30 ± 0.03 L min⁻¹ and sheath flows of 3.0 ± 0.3 L min⁻¹ resuling in the distribution of particles between 15 - 736 nm. Size distributions scans occurred

over 2.5 minute intervals PolyStyrene Latex (PSL) spheres were used to determine the sizing accuracy of both SMPS systems (± 2 %). The counting efficiency of the SMPS was determined by comparing the total number concentration of particles greater than 14 nm with the particle number concentration measured using the CPC TSI3772 and a scaling factor determined.  The SMPS concentrations were then scaled to the scaling factor.

 Figure 1 shows the time series of particle concentration as a function of diameter (particle size distribution) for SPS-I and

SPS-II.





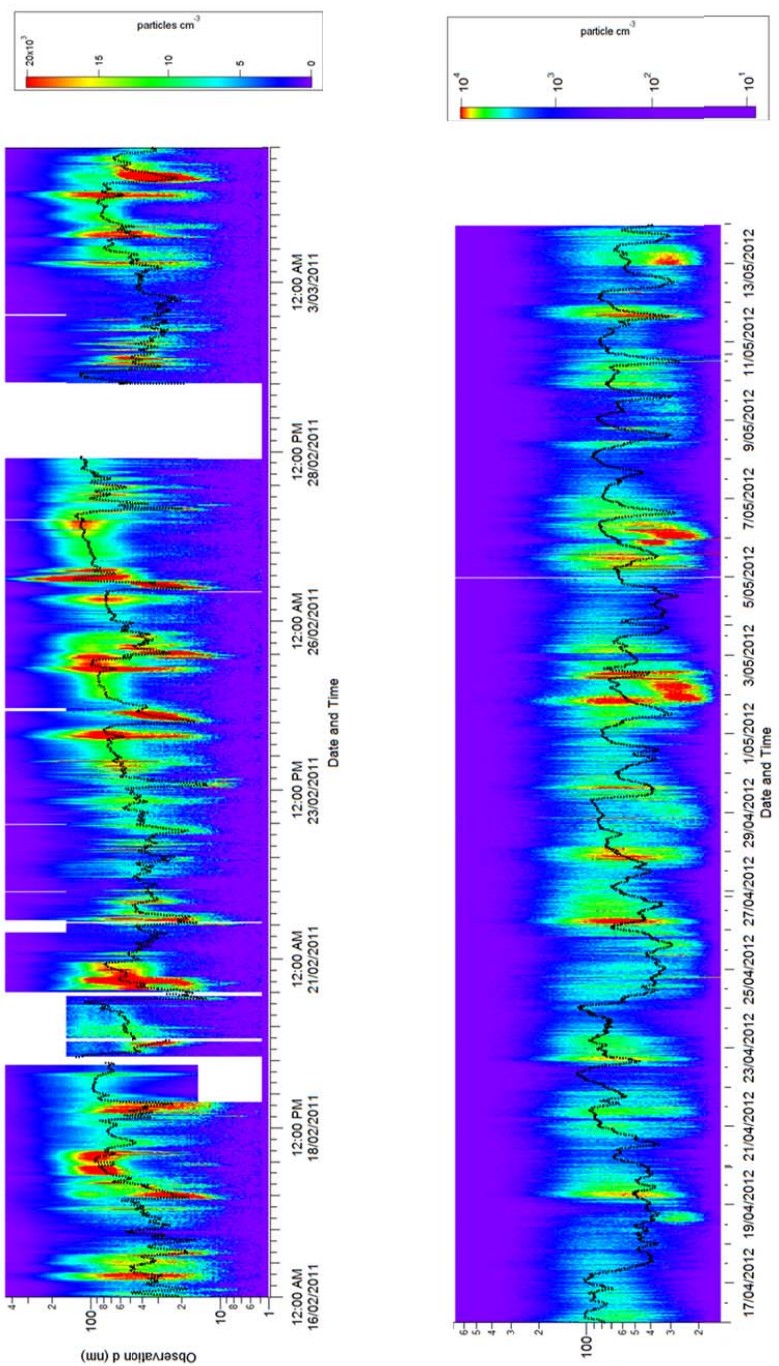

**Figure 1 Time series of aerosol size distribution for SPS-I (top panel) and SPS-II (bottom panel). The black dotted line on each plot is the mode diameter**





### 3.1.2 Aerosol scattering coefficient

During SPS-I and SPS-II light scattering was measured at 525 nm using an integrating nephelometer (Ecotech Aurora 1000G).
In this instrument, air is drawn into a chamber with a light beam at 525 nm and a photomultiplier detector set at right angles
to the light beam. Particles in the air scatter the light beam. The detector measures the scattered light beam in the forward and
backward direction. The nephelometer was operated according to the Australian Standard Method for integrated nephelometer
(AS/NZS 3580.12.1:2001). The inlet to the nephelometer was heated to ensure the relative humidity of the sample stream was

less than 40%. Daily zero air and span gas checks were carried out and the nephelometer was calibrated using $CO_2$ every three
months. Figure 2 shows the time series of aerosol scattering coefficient during SPS-I and SPS-II.



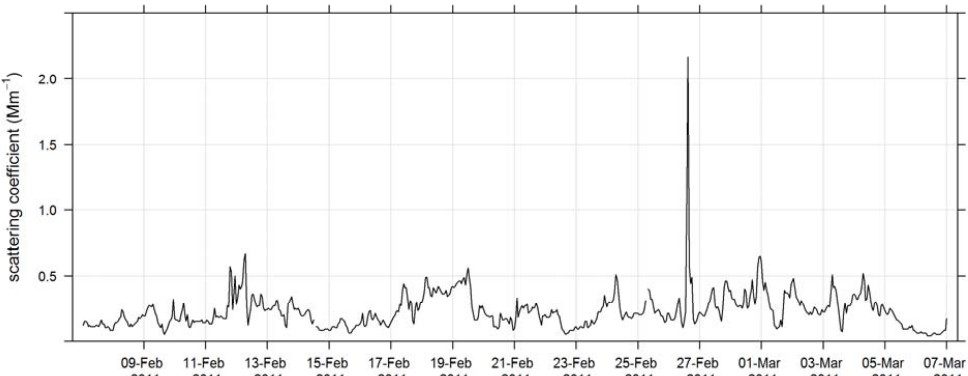

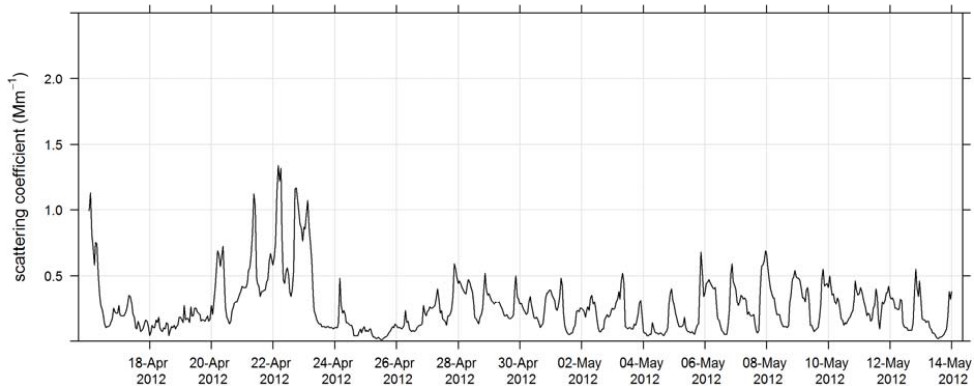

**Figure 2 Time series of hourly averaged aerosol scattering coefficients during SPS-I (top panel) and SPS-II (bottom panel)**

**3.1.3 PM$_{10}$**

During SPS-I and SPS-II the concentration of PM$_{10}$ was measured using a Tapered Element Oscillating Microbalance (Thermo TEOM1405). Air was drawn through a PM$_{10}$ impactor and a filter sitting on an oscillating microbalance. As mass loaded onto the filter, the frequency of oscillation changed and mass is recorded. The inlet to the TEOM was heated to 50 °C and the TEOM was operated according to Australian Standards for PM$_{10}$ continuous direct mass method using a tapered element

oscillating microbalance analyser (AS/NZ 3580.9.8-2008 ). Figure 3 shows the time series of PM$_{10}$ during SPS-I and SPS-II.



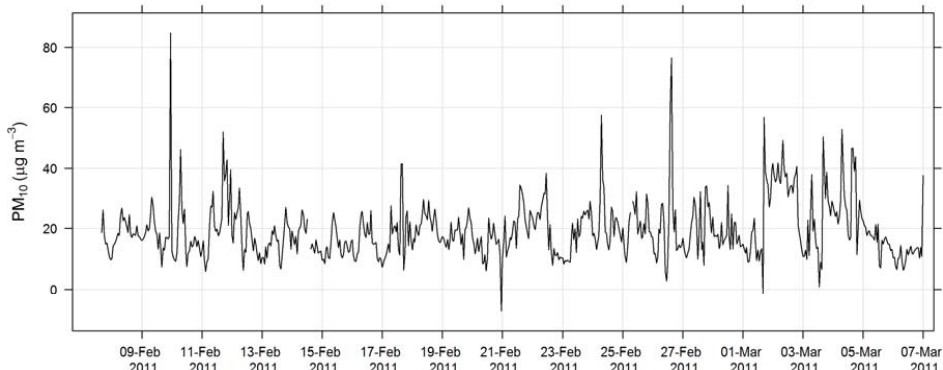

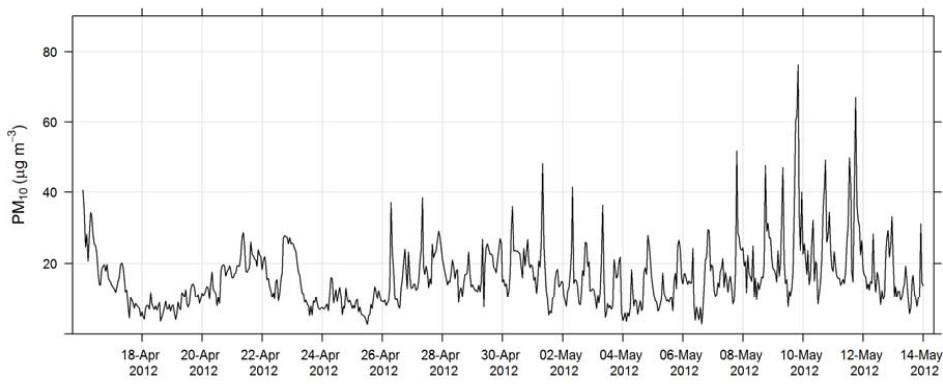

**Figure 3 Time series of hourly averaged PM$_{10}$ concentrations during SPS-I (top panel) and SPS-II (bottom panel)**

### 3.1.4 Proton transfer reaction mass spectrometer

Proton transfer reaction mass spectrometry (PTR-MS) is a chemical ionization mass spectrometry technique capable of quantifying volatile organic compounds (VOCs) in a gaseous sample at time resolutions down to a fraction of a second. The permanent constituents of air, oxygen, nitrogen, etc., are not detected. PTR-MS is suitable for the measurement of a range of atmospheric VOCs including aromatics, oxygenates, organo-sulphurs and terpenes.

The PTR-MS operates with the aid of a custom built auxiliary rack that regulates the flow of air in the sample inlet and controls

whether the PTR-MS is sampling ambient or zero air or calibration gas. During this study zero readings and calibrations against certified gas standards were performed on the PTR-MS several times per day. Four calibration standards were used during the




study, diluted to atmospheric concentrations using a set of mass flow controllers and a mixing chamber in the auxiliary rack. The PTR-MS was calibrated for: formaldehyde, acetaldehyde, acrolein, methacrolein, acetone, methyl ethyl ketone, methanol, ethyl acetate, benzene, xylene, trimethyl benzene, isoprene, a-pinene, 1,8 cineole, dimethyl sulphide, acetonitrile and the mono-

, di- and tri-chlorobenzenes. Only m/z that were detected above the method detection limit (MDL) greater than 25% of the time, and had peak to noise ratios greater than 5 (95th percentile/MDL) are reported. Further details are available in Galbally et al. (2007) and Dunne et al. (2012).

The time series of benzene, α-pinene and formaldehyde measured during SPS-I and SPS-II are shown in Figure 4 and Figure 5.

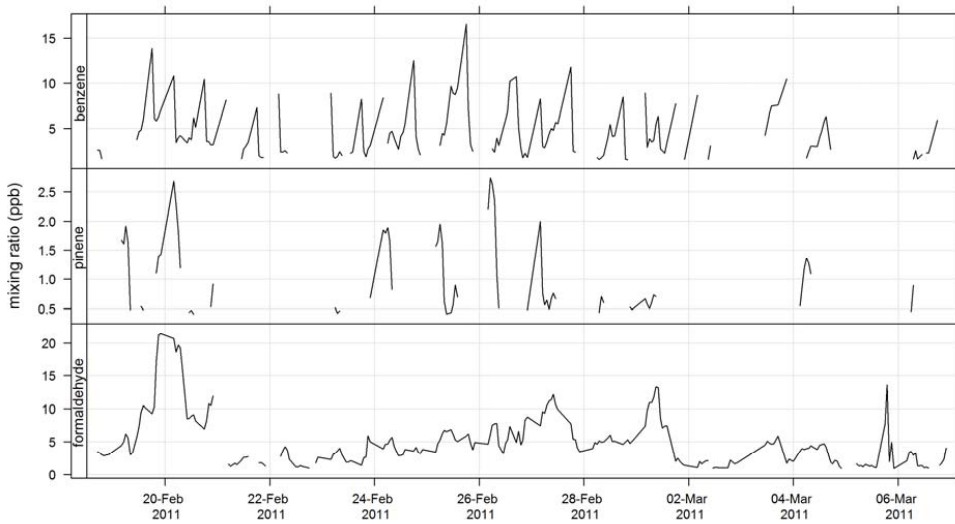


**Figure 4 Time series of ambient benzene, α-pinene and formaldehyde mixing ratios during SPS-I measured with the PTR-MS**



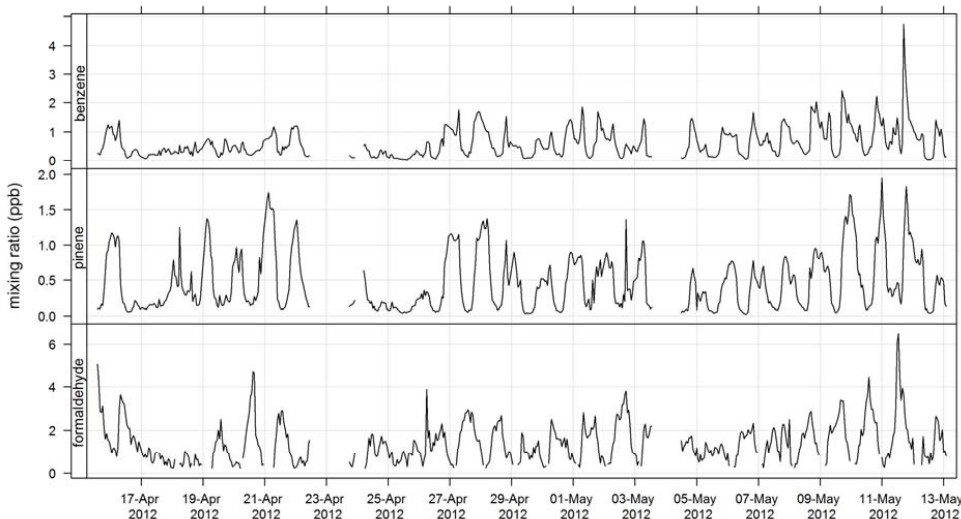

**Figure 5 Time series of ambient benzene, α-pinene and formaldehyde mixing ratios during SPS-II measured with the PTR-MS**

**3.1.5 Radon**

Radon concentration was measured using a dual flow-loop two-filter detection method (Whittlestone and Zahorowski, 1998; Chambers et al., 2014). The detector used for SPS-I and SPS-II was a 700 L model, which samplee at 40 L min$^{-1}$ from 2 m above ground level (45 minute response time, 40-50 mBq m$^{-3}$ lower detection limit). Operation followed the approach described by Chambers et al. (2011). An on-site calibration was carried out using a NIST traceable Pylon Ra-226 source (118.19 ±4% kBq), and instrumental background checks were carried out pre and post deployment.

In addition to the raw detector output, a time series of the atmospheric radon concentration was computed by deconvolving the detector output, thereby correcting for the slow detector response (Griffiths et al., 2016). The deconvolved time series has larger statistical uncertainty than the uncorrected detector output, but is a better representation of the atmospheric radon concentration during periods when it is changing rapidly (e.g. during the morning transition between nocturnal and convective boundary layers). Figure 6 shows the time series of radon measured during SPS-I and SPS-II.





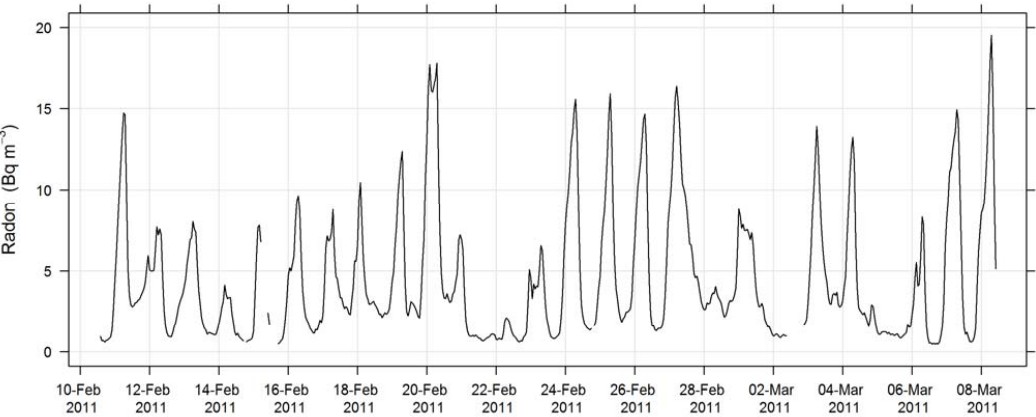


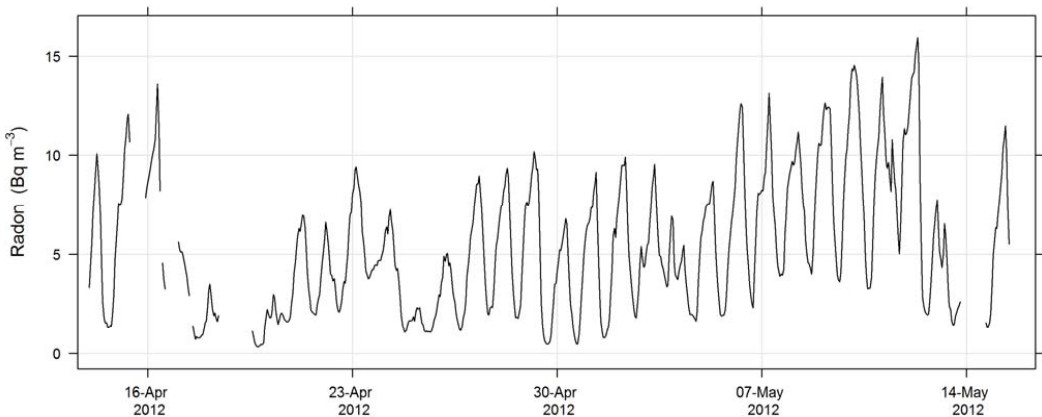

**Figure 6 Time series radon concentrations during SPS-I and SPS-II.**

### 3.1.6 Criteria Gases

Carbon Monoxide (CO) was measured using a nondispersive infrared CO analyser (Ecotech ML9830 CO trace gas analyser).

In this instrument, sample air is drawn into a cell where a beam of infrared light is passed through it to a photodetector. The

amount of light absorbed by CO in the sample is proportional to the number of molecules present, and the concentration of

CO is determined by comparing the intensity of light measured by the photodetector with a cell containing a reference gas.



The CO analyser was operated according to the Australian Standard method for the determination of CO by direct-reading instrumental method (AS/NZS 3580.7.1:2011).

Oxides of Nitrogen were measured using a chemiluminescent analyser (Ecotech EC9841 NOx trace gas analyser). In this instrument, nitric oxide (NO) in the sample air reacts with ozone (produced from an ultraviolet light) within a reaction chamber, producing chemiluminescence in the wavelength range 600–3000 nm. The concentration of NO is proportional to the light intensity measured by a photomultiplier tube. In a second sample stream, total nitrogen oxides (NOx) are reduced to NO using a selective converter. The concentration of nitrogen dioxide ($NO_2$) is assumed to be the difference between total NOx and NO.

The analyser was operated according to the Australian standard method for the determination of oxides of nitrogen by direct-reading instrumental method (AS/NZS 3580.5.1:2011)

Ozone ($O_3$) was measured using an ultraviolet spectrometer (Ecotech EC9810). In this instrument, a beam of ultraviolet light is passed through the sample air within a cell containing an ultraviolet detector. The amount of light absorbed in the sample is proportional to the number of $O_3$ molecules present and the decrease in light intensity determines the $O_3$ concentration in the

sample. The analyser was operated according to the Australian standard method for the determination of $O_3$ by direct-reading instrumental method (AS/NZS 3580.6.1:2011)

Sulfur dioxide ($SO_2$) was measured by pulsed fluorescence spectrophotometer (Ecotech EC9850). A stream of sample air is drawn through a cell where it is exposed to pulsed ultraviolet light, resulting in excitation of $SO_2$ molecules. These molecules subsequently fluoresce, by re-emitting light at a different wavelength. The intensity of the fluorescent light, as measured by a

photomultiplier tube, is proportional to the concentration of $SO_2$ in the sample air. The analyser was operated according to the Australian standard method for the determination of $SO_2$ by direct-reading instrumental method (AS/NZS 3580:5.1:2011, 2011;AS/NZS 3580:6.1:2011, 2011;AS/NZS 3580:7.1:2011, 2011)/NZS 3580.4.1:2008 ).

The time series for CO, NOx, $O_3$ and $SO_2$ for SPS-I and SPS-II are shown in Figure 7 and Figure 8.



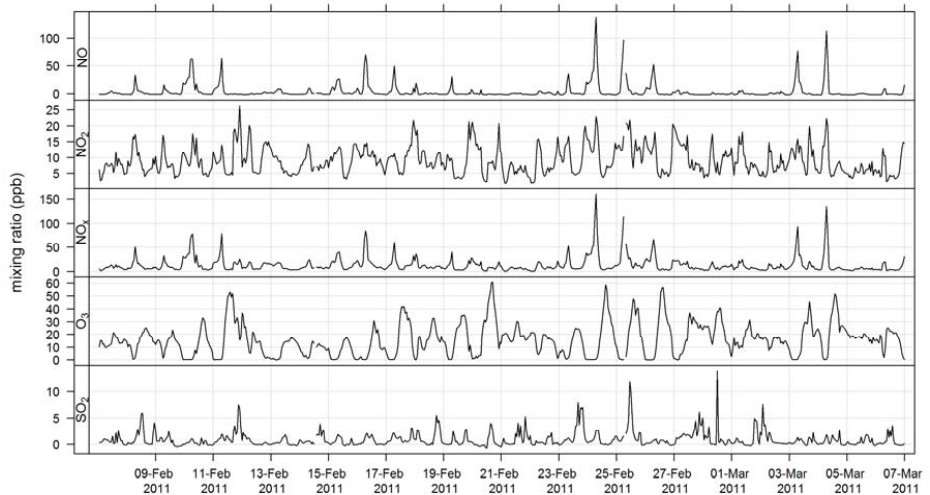

Figure 7 Time series of hourly averaged mixing ratios of criteria gases NO, NO₂, NOx, O₃ and SO₂ during SPS-I.

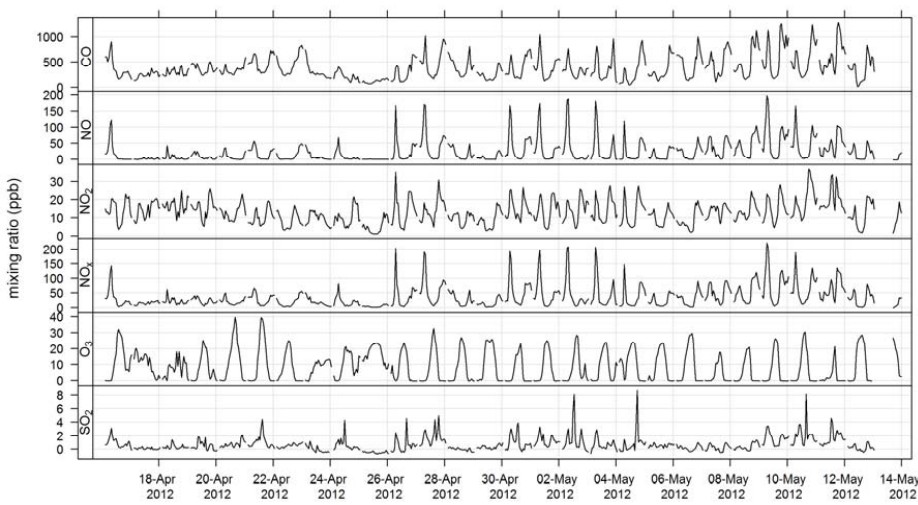

Figure 8 Time series of hourly averaged mixing ratios of criteria gases CO, NO, NO₂, NOx, O₃ and SO₂ during SPS-II



### 3.1.7 Meteorology

An ultrasonic ensor (Met-One MET505) was used to measure wind speed and wind direction. Temperature and relative humidity were measured using temperature and humidity probe (Vaisala HMP 155). Solar radiation was measured using a pyranometer (Middleton 8536). All instruments were sited and operated according to the Australian standard method for Meteorological monitoring for ambient air quality monitoring applications (AS/NZS 3580.14:2011).

The time series of temperature and relative humidity are shown for SPS-1 in Figure 9, in additon solar radiation is also shown
for SPS-II in Figure 10. The frequencies of wind speeds as a function of wind direction for SPS-I and SPS-II are shown in Figure 11.

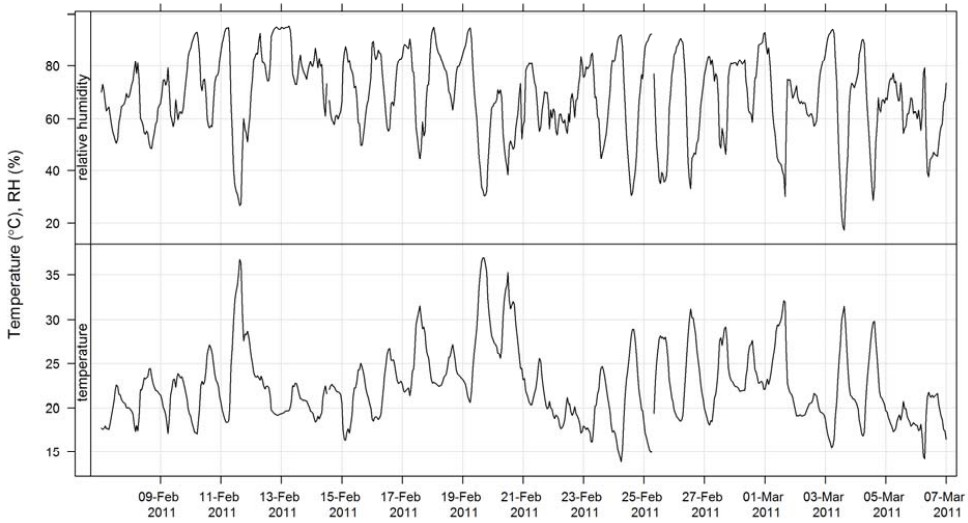

**Figure 9 Time series of ambient temperature and relative humidity during SPS-I.**





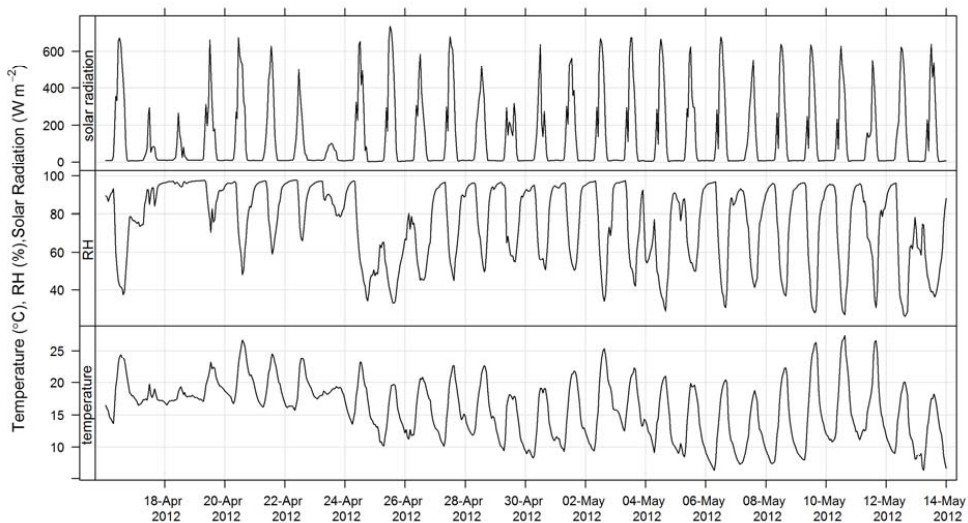

**Figure 10 Time series of ambient temperature, relative humidity and solar radiation during SPS-II**

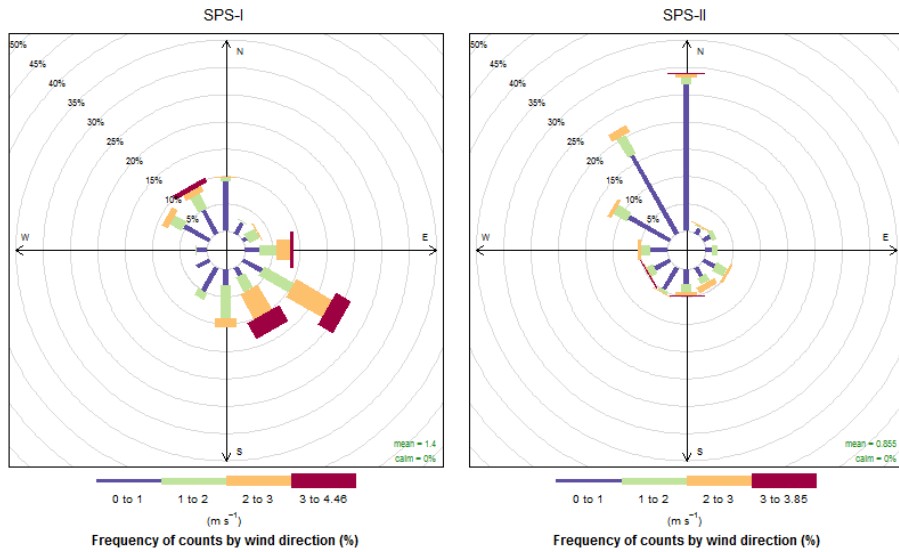

**Figure 11 Wind roses during SPS-I in 2011and SPS-II in 2012**



### 3.1.8 Lidar and boundary layer detection

A Leosphere ALS 450 lidar was used to estimate cloud base, cloud top (for optically thin clouds) and the height of the boundary layer. The lidar incorporated a 355 nm UV laser that scattered light in the column of air back to a receiver. Raw data have a spatial resolution of 15 m, temporal resolution of 30 s covering a range from about 200 m to 20 km. The physical basis for lidar remote sensing is described by Weitkamp (2005).

The conditions under which the lidar could determine the depth of the boundary layer included 1) the top of the boundary layer

being deeper than about 200 m, and 2) accompanied by a sudden decrease in aerosol concentration. These conditions were most often met during daylight hours with clear skies or some fair weather cumulus.

Two approaches were combined in order to filter out periods with ambiguous retrievals of the boundary layer depth. The first method was an automated method, called STRAT-2D (Haeffelin et al, 2012), used the Canny edge detection algorithm to detect discontinuities in the backscatter signal as a function of time and range. It is implemented in the STRAT analysis toolkit

(Morille et al., 2007). The second method was a manual technique, in which the boundary layer top was detected by visual identification of the inflection point in a plot of $\log(Sr^2) \sim r$, where S is the received backscatter and $r$ is the range from the lidar.

Based on the two estimates above, the boundary layer depth was computed by taking the average of the two estimates and assigning an uncertainty given by the range between the two estimates. Figure 12 shows the diurnal cycle in boundary layer

depth measured during SPS-II.

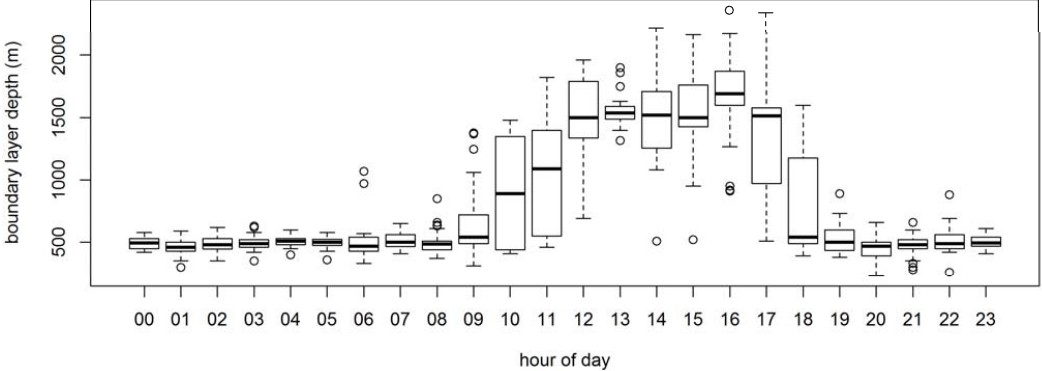



**Figure 12 Boundary layer depth as a function of hour of the day for SPS-II.**

### 3.2 Integrated measurements

**3.2.1 High Volume Sampler**

In both SPS-I and SPS-II aerosol samples were collected using an Ecotech 3000 high volume sampler with a $PM_{2.5}$ size-selective inlet (flow rate 67.8 $m^3$ $hr^{-1}$ controlled with a mass flow controller, ambient temperature and pressure monitored so that both the ambient volumetric and standard flow rates were determined). Quartz membrane filters (250 mm x 200 mm Pall tissuequartz p/n 7204 prebaked at 600°C for 4 hours to reduce adsorbed organic vapours) were used to collect samples and

were stored in a freezer within sealed containers before and after sampling.

Throughout the study field blank samples (5 for SPS-I and 9 for SPS-II) were collected by running a pre-baked filter the high volume sampler for 1 minute. Filter handling and analysis procedures were consistent for the field blanks and sample filters. In addition, to correct for sampling artefacts on the OC and EC concentrations, two filters were placed in the filter holder in sequence (front filter and back filter) for of 20 of the SPS-I samples. The adsorption of volatile gases onto the filter material

results in positive artefacts, while degassing of semi-volatile compounds from the collected aerosol on the front filter which may be then absorbed onto the back filter, results in negative artefacts arise.

The filters were analysed for soluble ions using the method described in Section 3.3.1 and for organic carbon, and the method described in Section 3.3.1 for elemental carbon.

**3.2.2 Low Volume sampler**

$PM_{2.5}$ samples were collected using a sampler from the ANSTO Aerosol Sampling Program which includes a $PM_{2.5}$ cyclone (flow rate 22 l $min^{-1}$). The cyclone is the same as that used in the US EPA IMPROVE network (http://vista.cira.colostate.edu/Improve/). Thin 25 mm stretched Teflon filter were used to collect samples coincidentally with the high volume sampler to allow comparison of data during SPS-II. Samples were analysed for elemental concentrations using the method described in Section 0.

**3.2.3 VOC and Carbonyls Sequencer**

The VOC and Carbonyls Sequencer is an automatic continuous air sampler for sampling of VOC and carbonyls simultaneously. It has two channels: one for VOC and the other one for Carbonyls. Each channel contains a sample inlet, 9 sampling ports, 4 solenoid valves and a sampling pump. A new sequencer was built for SPS-II that included a cooling system to keep the carbonyl tubes at 5-7 °C as well as extra sampling ports.

Samples were collected three times per day (05:00 – 10:00, 11:00 – 19:00 and 19:00 – 05:00) and during SPS-I a field blank (unopened tube) was collected each day. The new sequencer used in SPS-II incorporated extra sampling ports that were used to load extra sampling tubes that did not have any air sampled through them. These were then used as field blanks. The tubes



were analysed for VOC concentrations using the method described in Section 0 and for carbonyls using the method described in Section 0.

**3.2.4 Acid/Alkaline Gas sampler**

The acid/alkaline gas sampler drew air through a 3-stage 47mm filter pack at an ambient flow rate of $10 \, l \, min^{-1}$. The first stage of the 3 stage filter pack contained a Teflon filter (Millipore fluoropore p/n FALP04700) to remove particles from the air stream, the second stage contained a sodium hydroxide coated quartz filter (Pall tissuequartz p/n 7202) to trap acidic gases and the final stage contained a citric acid coated quartz filter to trap alkaline gases. The filters were extracted in de-ionized water and analysed for soluble ion concentrations using the method described in Section 0.

**3.3  Analysis methods**

**3.3.1  Ion chromatography**

Suppressed ion chromatography (IC) and high-performance anion-exchange chromatography with pulsed amperometric detection (HPAEC-PAD) were used to measure water soluble ions and anhydrous sugars including levoglucosan (respectively) on a $6.25 \, cm^2$ a portion of each quartz high volume sampler filter. De-ionized water (10 ml of $18.2 \, m\Omega$) was used to extract the quartz filter portions which were then preserved using 0.1 ml of chloroform. The acid and alkaline gas filter samples were also analysed by IC and the 47 mm filters were extracted in 3 ml of $18.2 \, m\Omega$ de-ionized water and preserved with 0.03 ml of chloroform.

A Dionex ICS-3000 ion chromatograph was used to determine soluble ion (anion and cation) concentrations. The system included a a Dionex AS17c analytical column (2 x 250 mm), an ASRS-300 suppressor and a gradient eluent of 0.75 mM to 35 mM potassium hydroxide to separate the anions, and aDionex CS12a column (2 x 250 mm), a CSRS-300 suppressor and an isocratic eluent of 20 mM methanesulfonic acid to separate the cations.  The species analysed were

- Chloride ($Cl^-$)
- Nitrate ($NO_3^-$)
- Sulphate ($SO_4^{2-}$)
- Oxalate ($C_2O_4^-$)
- Formate ($HCOO^-$)
- Acetate ($CH_3COO^-$)
- Phosphate ($PO_4^{3-}$)
- Methanosulfonate ($MSA^-$)

- Sodium ($Na^+$)
- Ammonium ($NH_4^+$)
- Magnesium ($Mg^{2+}$)
- Calcium ($Ca^{2+}$)
- Potassium ($K^+$)

The time series for $Mg^{2+}$, $Cl^-$ $NH_4^+$ and $SO_4^{2-}$ during SPS-I are shown in Figure 13 and those during SPS-II in Figure 14.





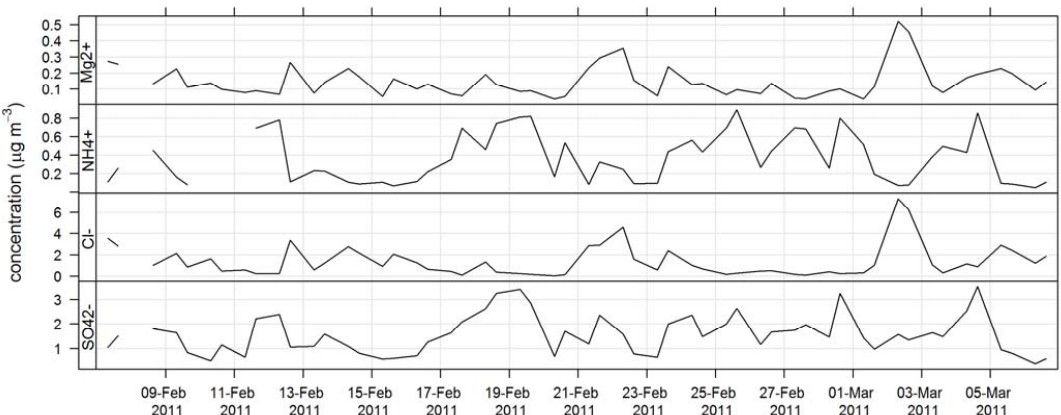

**Figure 13 Time series of $Mg^{2+}$, $Cl^-$ $NH_4^+$ and $SO_4^{2-}$ during SPS-I in 2011**

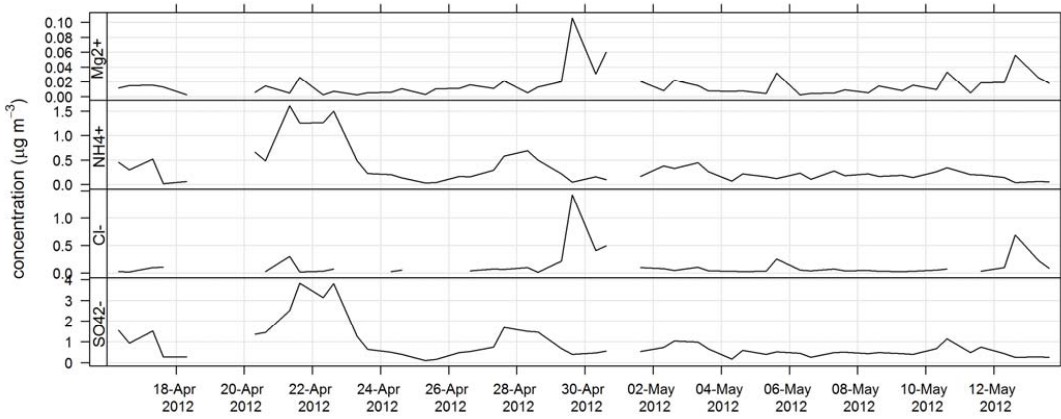

**Figure 14 Time series of $Mg^{2+}$, $Cl^-$ $NH_4^+$ and $SO_4^{2-}$ during SPS-II in 2012**

An HPAEC-PAD with a Dionex ICS-3000 chromatograph with electrochemical detection was used to determine anhydrous sugar concentrations. The system was operated in the integrating (pulsed) amperometric mode using the carbohydrate 280 (standard quad) waveform and utilizing disposable gold electrodes. A Dionex CarboPac MA 1 analytical column (4 x 250mm) with a gradient eluent of 300 mM to 550 mM sodium hydroxide was used to separate the anhydrous sugars (Iinuma et al.,



2009). The species analysed were levoglucosan (C$_6$H$_{10}$O$_5$, an anhydrous sugar - woodsmoke tracer) and Mannosan (C$_6$H$_{10}$O$_5$, an anhydrous sugar - woodsmoke tracer). The time series of levoglucosan during SPS-I and SPS-II are shown in Figure 15.

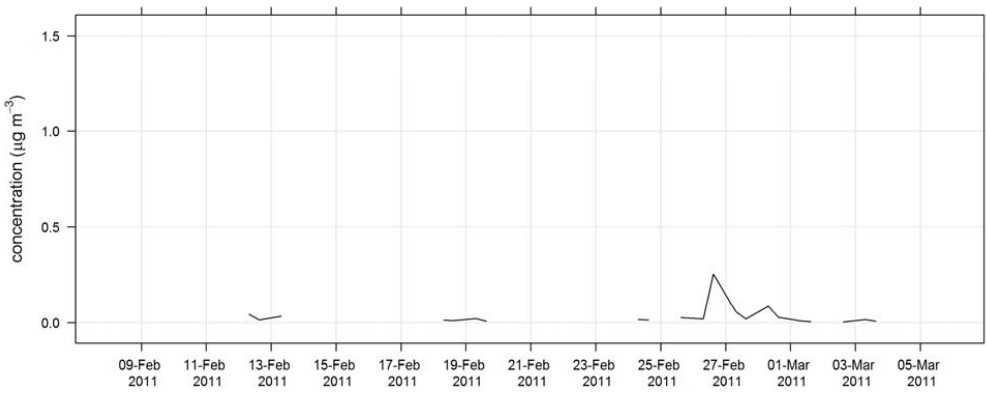

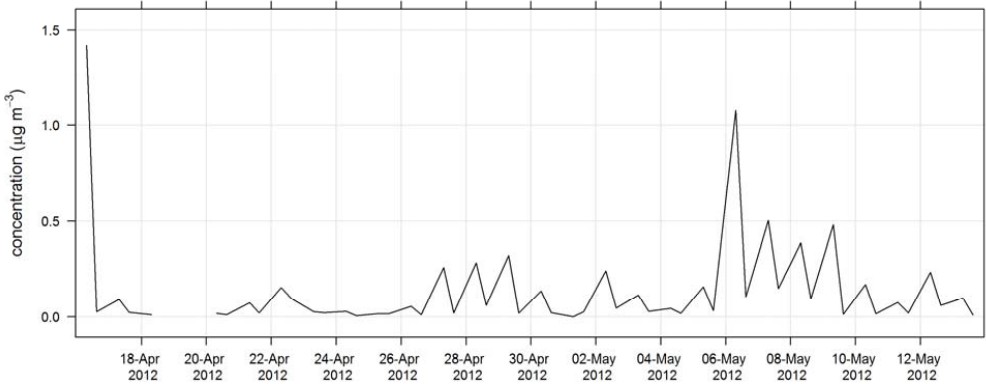


**Figure 15 Time series of levoglucosan concentrations during SPS-I (top panel) and SPS-II (bottom panel)**

**3.3.2 Ion beam analysis**

Nuclear ion beam analysis (IBA) techniques suing the non-destructively on the ANSTO STAR 2MV accelerator was used to determine the concetration of elements on the 25 mm Teflon filters collected by the low volume sampler. Analysis of

aluminium to lead was carried out using Proton induced X-ray emission (PIXE see Cohen 1993 for details); analysis of light



elements such as fluorine and sodium was carried out by Proton induced gamma-ray emission (PIGE see 1998 for details) and analysis of hydrogen was carried out using Proton elastic scattering analysis (PESA see Cohen 1996 for details). Key
The elements determined were:

- Hydrogen (H)
- Sodium (Na)
- Aluminium (Al)
- Silicon (Si)
- Phosphorous (P)
- Sulfur (S)
- Chlorine (Cl)
- Potassium (K)
- Calcium (Ca)
- Titanium (Ti)

- Vanadium (V)
- Chromium (Cr)
- Manganese (Mn)
- Iron (Fe)
- Cobolt (Co)
- Nickel (Ni)
- Copper (Cu)
- Zinc (Zn)
- Bromine (Br)
- Lead (Pb)

The time series of Al and Si for SPS-II are shown in Figure 16.


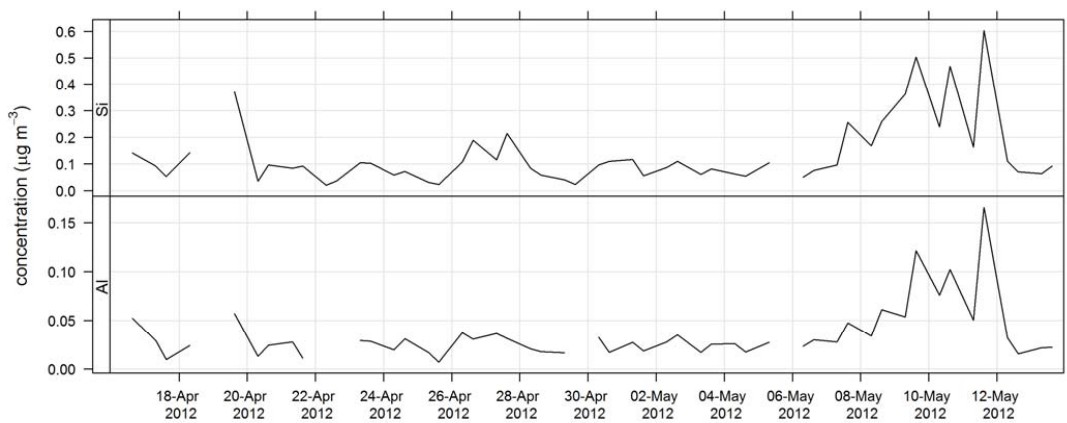

**Figure 16 Time series of Al and Si during SPS-II**

### 3.3.3 Organic carbon and Elemental carbon analysis

A DRI Model 2001A Thermal-Optical Carbon Analyzer was used to determine the concentration of elemental carbon (EC)
and organic carbon (OC) on a portion of the quartz filters collected using $PM_{2.5}$ high volume sampler. The IMPROVE-A temperature protocol (Chow et al., 2007) was employed and included using laser reflectance to correct for charring. Before analysis the oven was baked to 910°C for 10 minutes to remove residual carbon and system blank levels are then tested until $< 0.20 \ \mu g \ C \ cm^{-2}$ was reported (with repeat oven baking if necessary). Calibration checks were peformed twice daily to monitor




possible catalyst degeneration. The analyser is reported to measure carbon concentrations between 0.05 – 750 µg C cm$^{-2}$, with

uncertainties in OC and EC of ± 10%.

Four OC fractions at four non-oxidizing heat ramps (OC1 =140°C, OC2 = 280°C, OC3 = 480°C, OC4 = 580°C) and three EC

fractions at three oxidizing heat ramps (EC1 = 580°C, EC2 = 740°C, EC3 = 840°C) are measured in the IMPROVE-A carbon

method. The sum of the different OC fractions and the OCpyro (the OC that was pyrolized which was measured from the

reflectance of the filter) determined total OC.  The sum of the EC fractions minus OCpyro determined total EC.

The time series for OC and EC during SPS-I are shown in Figure 17 and the time series for OC and EC during SPS-II are

shown in Figure 18.

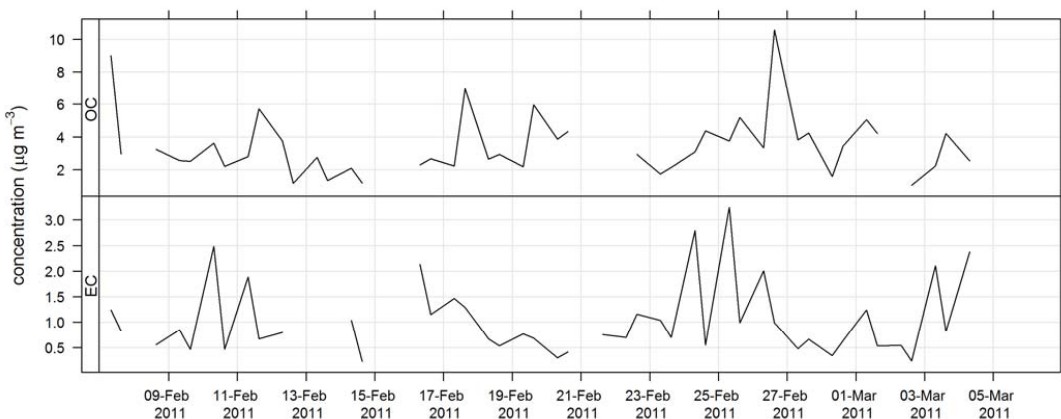

**Figure 17 Time series of OC and EC during SPS-I in 2011**





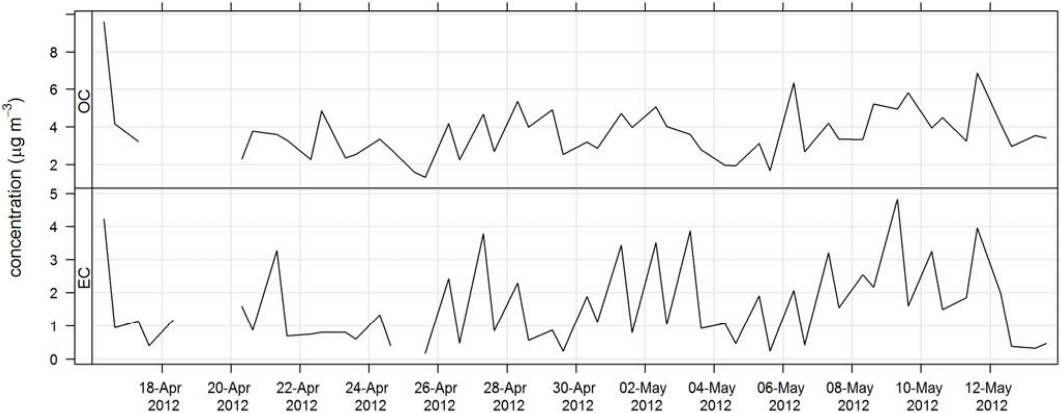


**Figure 18 Time series of OC and EC during SPS-II in 2012**

### 3.3.4 Carbonyls analysis

Carbonyls were collected by the sequencer onto cartridges (Supelco LpDNPH S10 p/n 21014) containing high-purity silica adsorbent coated with 2,4-dinitrophenylhydrazine (DNPH), where they were converted to the hydrazone derivatives. Samples
were refrigerated immediately after sampling until analysis. The derivatives were extracted from the cartridge in 2.5 mL of acetonitrile and analysed by high performance liquid chromatography (HPLC) with diode array detection (DAD). The DAD enables the absorption spectra of each peak to be determined. The difference in the spectra highlights which peaks in the chromatograms are mono- or dicarbonyl DNPH derivatives and, along with retention times, allows the identification of the dicarbonyls glyoxal and methylglyoxal. Further details of this method can be found in Lawson et al. (2015).
The time series of methylglyoxal and formaldehydefor SPS-I is shown in Figure 19 and time series of methylglyoxal and formaldehyde during SPS-II are shown in Figure 20.



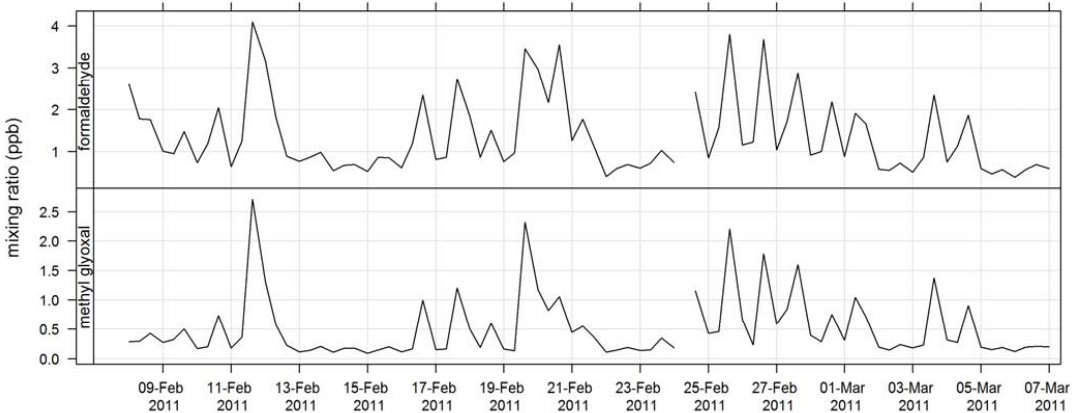

**Figure 19 Time series of ambient formaldehyde and methylglyoxal mixing ratios mixing ratios during SPS-I.**

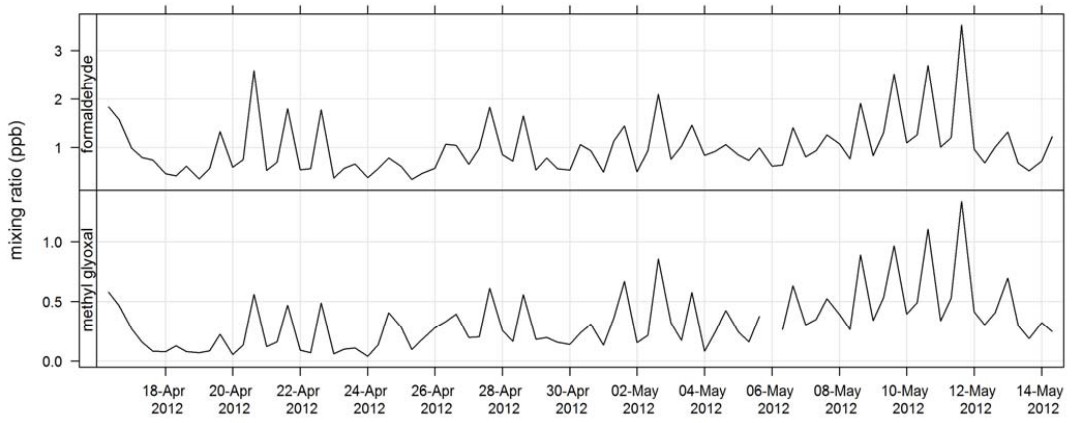

**Figure 20 Time series of ambient formaldehyde and methylglyoxal mixing ratios concentrations during SPS-II.**

### 3.3.5 Volatile organic compounds analysis

An automatic Volatile organic compound (VOC) sampler was used collect VOC samples by actively drawing air through two adsorbent tubes in series (Markes Carbograph 1TD / Carbopack X) which were then analysed by a PerkinElmer TurboMatrix™ 650 ATD (Automated Thermal Desorber) and a Hewlett Packard 6890A gas chromatograph (GC) equipped with a Flame





Ionization Detector (FID) and a Mass Selective Detector (MSD). Calibration was via certified BTEX (benzene, toluene, ethylbenzene and xylenes), TO 15/17, terpenes, alcohols and PAM gas standards (.Cheng et al. 2016). The method of AT (adsorbent tube) VOC sampling and analysis in this study was compatible with ISO16017-1:2000 (ISO 2000) and according to USEPA Compendium method TO-17 (USEPA TO-17). The time series for total alkane, aromatic and terpene concentrations for SPS-I are shown in Figure 21 and for SPS-II in Figure 22.

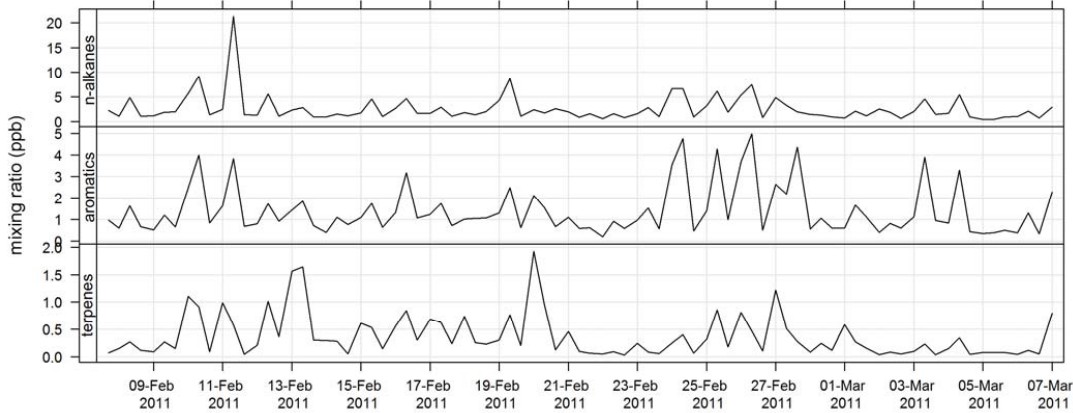


**Figure 21 Time series of total alkane, total aromatics and total terpene mixing ratios during SPS-I in 2011measured on absorbent tubes.**

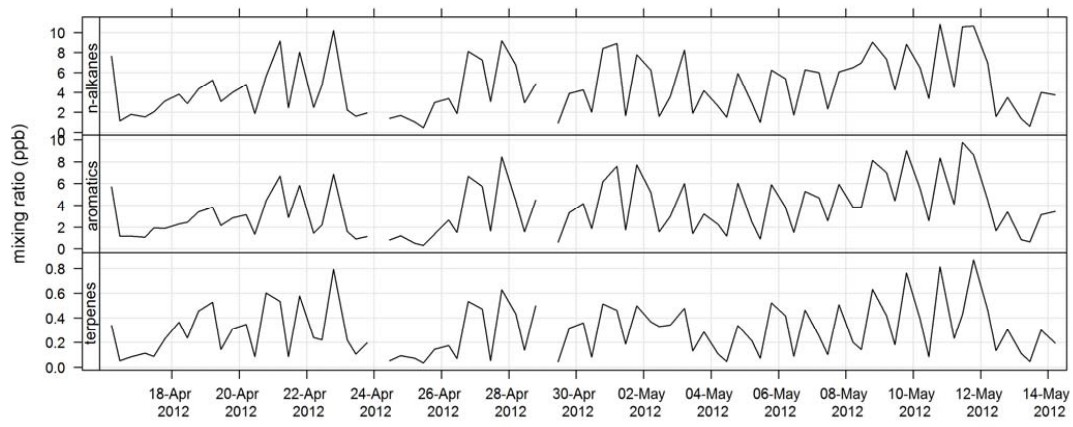

**Figure 22 Time series of total alkane, total aromatics and total terpene mixing ratios during SPS-II in 2012measured on absorbent tubes.**



**4. Aerosol composition**

The factors that determine the composition of particles are the source of the particles (or precursor gases) and subsequent transformations that occur in the atmosphere or within the particles themselves. As such the sources of particles may be inferred from the chemical composition of the particle samples. A detailed analysis of the aerosol composition data is beyond the scope of a paper in this journal. Instead, presented here are the data for some species that can be used as markers for different aerosol sources.

Table 2 lists the markers that can be used to trace different aerosol sources. In some instances, a chemical species is a unique tracer for a source. For example, levoglucosan is a unique tracer for biomass burning (Simoneit et al., 1999; Simoneit, 2002). The time series of levoglucosan for both sampling periods is shown in Figure 15. Concentrations were generally greater in SPS-II than SPS-I, indicating more biomass burning (most likely woodheaters for domestic heating during autumn in Sydney in SPS-II).

In addition the ratios of different species may provide information about a particle source. For example a sea salts source may ne indicated by a [$Na^+$/$Mg^{2+}$] ratio close to 8.3 (Millero et a., 2008), and an Australian crustal dust source may be indicated by a [Si/Al] ratio of close to 3.08 (Radhi et al., 2010). Figure 23 shows the relationship between $Na^+$ and $Mg^{2+}$ for SPS-I and SPS-II, suggesting that the ratio of these species is close to that of sea-salt. Figure 24 shows the relationship between Si and Al for SPS-II. The slope of 4 shown in the regression line is similar to that measured by Radhi et al. (2010), indicative of

Australian dusts.

**Table 2. Sources and their indicator species**

| Source | Indicator Species |
|---|---|
| **Soil** | Non sea salt Calcium (SPS-I)<br><br>Silicon, Iron, Aluminium , Titanium (SPS-II) |
| **Organic Matter (OM) – Vehicles, Industry, Biomass Burning (BB), secondary organic aerosol (SOA)** | Organic Carbon |
| **Elemental Carbon (EC)- Vehicles, Industry, BB** | Elemental Carbon |
| **Sea Salt** | Sodium, Chloride, Magnesium |
| **Secondary Inorganic Aerosol (SIA)** | Non sea salt Sulfate , Ammonium  Nitrate |
| **Biomass Burning (BB)** | Levoglucosan |





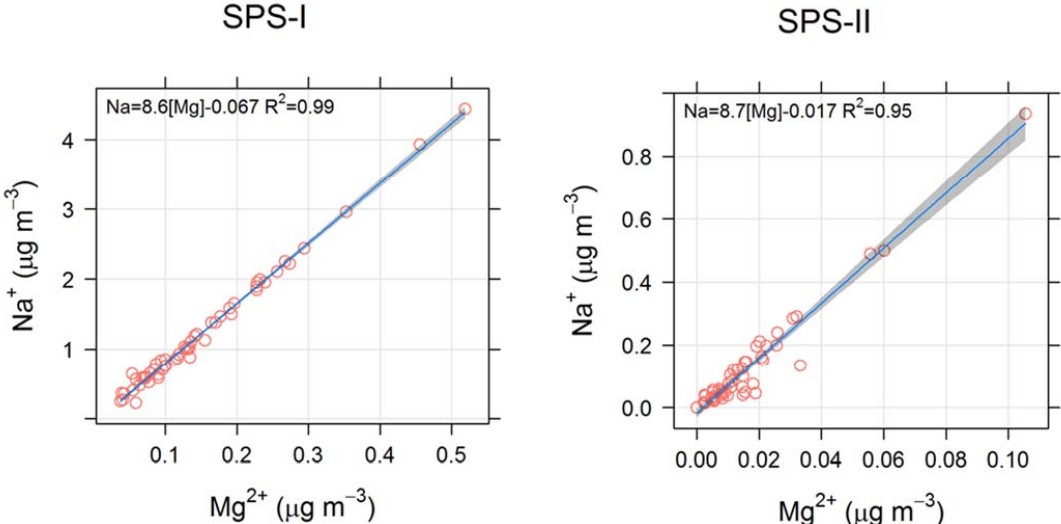

**Figure 23 Scatter plot of Mg and Na for SPS-1 (left) and SPS-II (right).**

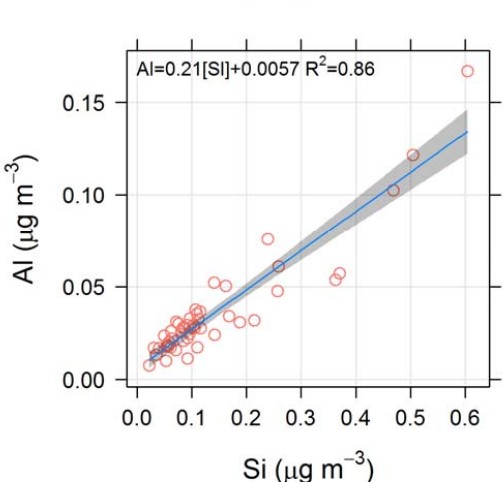

**Figure 24 Figure 25  Scatter plot of Al and Si for SPS-II.**



Many compounds, however, may be derived from more than one source (e.g., EC can be emitted by vehicle emissions,
industrial emissions and biomass burning) and when sufficient sample numbers allow (generally more than 100) receptor
modelling methodologies can be used to apportion sources to the aerosol loadings (Norris et al., 2008). During both SPS-I
and SPS-II 30 samples were collected in the mornings and 30 in the afternoons. Hence with only 60 samples for each sampling
period, we are restricted to a qualitative and rudimentary assessment of aerosol sources utilising information on relationships
between some key marker species and the timing of their occurrence.

The time series for EC, OC, $SO_4^{2-}$, $Mg^{2+}$ and $Ca^{2+}$ are shown above. The average concentrations for SPS-I and SPS-II for each
of these species are shown in Figure 26. A marker for sea-salt, $Mg^{2+}$ and a marker for soil, non sea salt $Ca^{2+}$, show higher
concentrations during SPS-I (summer). Higher non sea salt sulfate concentrations, a marker for secondary aerosol during
summer may indicate greater secondary aerosol production during summer. Levoglucosan (the marker for woodsmoke) and
EC show highest concentrations during the SPS-II (autumn). The average OC concentration is not significantly different
between SPS-I and SPS-II.

Higher sea-salt and soil marker species in summer than autumn may be due to higher wind speeds observed during SPS-I since
both sea-salt and dust are mechanically produced aerosol. Figure 11 also showed there to be a greater recent oceanic fetch in
summer; and during summer soils in rural regions are drier, and covered with less vegetation, so therefore more mobile. Higher
secondary aerosol marker species in summer may indicate more photochemical aerosol production in summer, while higher
biomass burning marker species in autumn may represent a greater contributions from woodheaters to the aerosol loading
during this time of the year. Converse to the higher wind speeds during summer, the lower windspeeds during autumn are also
conducive to the build-up of pollutants during autumn which will also influence the concentrations of EC and levoglucosan
during autumn. As noted above, a quantitative assessment of aerosol sources influencing the airshed during SPS-I and SPS-II
could be carried out using a receptor modelling approach if more samples had been collected.




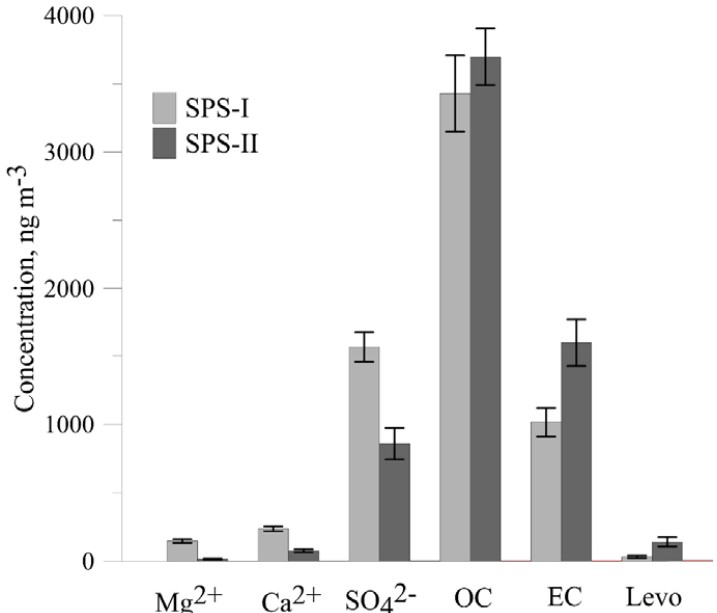

**Figure 26 Comparison of average concentrations of Mg (a marker for sea-salt), Ca (a marker for soil), SO₄ (a marker for secondary aerosol), OC, EC and levo = levoglucosan (biomass burning marker) during SPS-I and SPS-II. Error bars represent standard error (standard deviation/square root of number of observations). Mg, Ca, and SO₄ are significantly greater during SPS-I (p <<0.05), OC is not significantly different between SPS-I and SPS-II (p=0.4), EC and levoglucosan are significantly greater during SPS-II (p= 0.003 EC and p=0.004 levoglucosan).**

### 5 Data set repository and description

The data sets for both SPS-I and SPS-II are stored on the Commonwealth Scientific and Industrial Research Organisation (CSIRO) data access portal. The SPS-I data set is available at Keywood et al. (2016a) http://doi.org/10.4225/08/57903B83D6A5D and the SPS-II data set is available at Keywood et al. (2016b) http://doi.org/10.4225/08/5791B5528BD63.

### 6 Acknowledgements

Descriptive and statistical analyses were carried out using R statistical analysis version 3.5.1 (R Core Team, 2016). The main package used in R statistical analysis is known as "openair" version 2.4.2 (Carslaw and Ropkins, 2012). This work was funded by NSW Office of Environment and Heritage and CSIRO.




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
