# Peer review of "Comprehensive aerosol and gas data set from the Sydney Particle Study"

_Earth System Science Data, 2019_

## Referee Comment (RC1) · Guy Coulson (Referee) · 6 Jun 2019

This paper presents itself as the metadata for two datasets from the Sydney Particle Study in 2011/12 available on the CSIRO servers. The two datasets are easily accessible following the links in the paper and easy to download.

The paper itself is well written and is a good overview of the data. The descriptions of the data and accompanying plots are sufficiently detailed to inform potential users of the type and amount of data available.

There is no information on any quality assurance processing, which would add to confidence in the data.

There are a few typos, for example, NOy in table 1 (page 5) rather than NOx but nothing

serious - a final, good proofread required.

---

## Referee Comment (RC2) · Anonymous Referee #2 · 17 Jun 2019

The Comprehensive aerosol and gas data set from the Sydney Particle Study presents two sets of data collected in two seasons at the western Sydney location. The large number of variables and the easy accessibility of data will help various studies and the manuscript can be considered for publication.

The text describes the data in detail, however, some improvements could help readers:

In the text: L 70, table I should be described in more detail and in particular the content of the columns 'resolution' and 'reported resolution'.

Table I: could be organized with the variables in line according to the text flow.

Figures: when possible, for a better comparison (see fig.1 for example),. the same vertical scale should be used for both data sets.

---

## Author Comment (AC1) · 4 Aug 2019

We would like to thank Referee 1 for his time taken to review this manuscript and for his comments which will result improvements to the manuscript. We have addressed the comments below. Referee 1 comment This paper presents itself as the metadata for two datasets from the Sydney Particle Study in 2011/12 available on the CSIRO servers. The two datasets are easily accessible following the links in the paper and easy to download. The paper itself is well written and is a good overview of the data. The descriptions of the data and accompanying plots are sufficiently detailed to inform potential users of the type and amount of data available. There is no information on any quality assurance processing, which would add to confidence in the data. Our response The manuscript does include descriptions of the calibration processes

used to ensure high quality data in the method description for each parameter and we have assigned an uncertainty associated with the check were possible. For example, the text for the PTR-MS description is reproduced below The PTR-MS operates with the aid of a custom built auxiliary rack that regulates the flow of air in the sample inlet and controls whether the PTR-MS is sampling ambient or zero air or calibration gas. During this study zero readings and calibrations against certified gas standards were performed on the PTR-MS several times per day. Four calibration standards were used during the study, diluted to atmospheric concentrations using a set of mass flow controllers and a mixing chamber in the auxiliary rack. The PTR-MS was calibrated for: formaldehyde, acetaldehyde, acrolein, methacrolein, acetone, methyl ethyl ketone, methanol, ethyl acetate, benzene, xylene, trimethyl benzene, isoprene, a-pinene, 1,8 cineole, dimethyl sulphide, acetonitrile and the mono-, di- and tri-chlorobenzenes. Only m/z that were detected above the method detection limit (MDL) greater than 25% of the time and had peak to noise ratios greater than 5 (95th percentile/MDL) are reported. Further details are available in Galbally et al. (2007) and Dunne et al. (2012). In addition, where possible we operated the instruments following an Australian or international standard method which are put in place to ensure high quality data. The standards followed and reported in the manuscript are 1. AS 3580.4.1-2008: Methods of sampling and analysis of ambient air Determination of sulfur dioxide - Direct reading instrumental method, 2008. 2. AS/NZ 3580.9.8-2008: Determination of suspended particulate matter—PM10 continuous direct mass method using a tapered element oscillating microbalance analyser, 2008. 3. AS/NZS 3580.12.1:2001: Methods for sampling and analysis of ambient air - Determination of light scattering - Integrating nephelometer method, 2001. 4. AS/NZS 3580.14:2011: Methods for sampling and analysis of ambient air - Part 14: Meteorological monitoring for ambient air quality monitoring applications, 2011. 5. AS/NZS 3580.5.1:2011: Methods of Sampling and Analysis of Ambient Air – Determination of oxides of nitrogen- Direct reading instrumental method., 2011. 6. AS/NZS 3580.6.1:2011: Methods of Sampling and Analysis of Ambient Air – Determination of ozone- Direct reading instrumental method., 2011.

[Figure]

7. AS/NZS 3580:7.1:2011: Methods of Sampling and Analysis of Ambient Air – Determination of carbon monoxide- Direct reading instrumental method., 2011. Referee 1 comment There are a few typos, for example, NOy in table 1 (page5) rather than NOx but nothing serious - a final, good proofread required Our response We agree and have carried out a thorough proof read of the manuscript. We have corrected a number of minor mistakes in the text and in the reference list.

---

## Author Comment (AC2) · 4 Aug 2019

We would like to thank Referee 2 for their time taken to review this manuscript and for their comments which will result improvements to the manuscript. We have addressed the comments below. REFEREE 2 COMMENT The Comprehensive aerosol and gas data set from the Sydney Particle Study presents two sets of data collected in two seasons at the western Sydney location. The large number of variables and the easy accessibility of data will help various studies and the manuscript can be considered for publication. The text describes the data in detail, however, some improvements could help readers: In the text: L 70, table I should be described in more detail and in particular the content of the columns 'resolution' and 'reported resolution'. OUR RESPONSE We have changed the paragraph describing Table 1 to "Table 1 provides a summary
of the parameter measured and the instrument used to measure the parameter. The frequency at which the measurement of each parameter was made is also listed in the table ranging from continuous to one measurement every few minutes to the collection of a sample over several hours (integrated). The frequency at which the data are reported is also included in Table 1 as well as the units the measurements are reported in and whether the measurements were made during SPS-I, SPS-II or during both periods". We have also changed the headings from resolution and reported resolution to frequency collected and frequency reported and have added a description of what these terms mean in the table heading (reproduced below). Table 1. Measurements made at Westmead during SPS-I and SPS-II along with the instrument or analytical technique employed, the measurement and reporting resolution, and the measurement units. Frequency of measurement is the frequency with which the data are collected. Frequency reported is the frequency at which the data are reported (may be an average of the frequency of measurement). REFEREE 2 COMMENT Table I: could be organized with the variables in line according to the text flow. OUR RESPONSE While we acknowledge the benefit of organising the variables in Table 1 in line with the flow of the text, we believe that the benefit of the current order to the reader is greater. In particular, the text order discusses continuous measurements followed by integrated measurements. Included in the integrated measurement text are descriptions of the analytical procedures used to analyse the samples collected. We feel this provides a logical flow in the text. If we were to reorganise the variables in Table 1 to reflect the flow of the text, we would see a number of similar variables split e.g. VOCs across the table. Instead, we feel that ordering the table around variables is more useful to the reader. For example the reader can easily determine all the methods that were used to measure VOCs at a glance rather than having to scroll across two pages of the table. Hence, we have chosen not to adopt this suggestion. REFEREE 2 COMMENT Figures: when possible, for a better comparison (see fig.1 for example), the same vertical scale should be used for both data sets. OUR RESPONSE We have amended the vertical scales on the plots of Figure 1 to make them consistent